# Food taboos and preferences among adolescent girls, pregnant women, breastfeeding mothers, and children aged 6–23 months in Mainland Tanzania: A qualitative study

Aika Lekey[1], Ray M. Masumo[1]*, Theresia Jumbe[2], Mangi Ezekiel[3], Zahara Daudi[1], Nangida J. Mchome[1], Glory David[1], Winfrida Onesmo[3], Germana H. Leyna[1,3]

1 Tanzania Food and Nutrition Centre (TFNC), Dar es Salaam, Tanzania, 2 Sokoine University of Agriculture (SUA), Morogoro, Tanzania, 3 Muhimbili University of Health and Allied Sciences (MUHAS), Dar es Salaam, Tanzania

* rmasumo@yahoo.com

**Data Availability Statement:** All the data for this study is available here and can be accessed without restriction; it belongs to the authors, URL: https://

## Abstract

Malnutrition is a serious public health problem and has long-lasting economic consequences for individuals and families and, in turn, affects the economic growth of the country. Understanding food taboos and individual preferences for food items is critical to the development of effective nutritional programs and educational messages. The present study aimed to explore food taboos and preferences in food items among breastfeeding mothers, pregnant women, adolescent girls, and their young children aged 6–23 months old. This is a qualitative cross-sectional study employing ethnography. A multistage sampling technique was used to select one region from the seven zones in mainland Tanzania. In each region, we purposively selected one rural ward and one urban ward. We conducted 25 focus group discussions with a total of 208 participants. We also conducted 42 in-depth interviews with nutrition officers, community health workers, religious leaders, influential persons, representatives of civil society organisations, and community leaders in the respective areas. We thematically coded the data and analyzed the narrative. Food taboos and individual preferences in food items continue to be practiced in Tanzania's Mainland despite efforts to educate people on healthy diets. In some regions of Tanzania's Mainland, pregnant women and breastfeeding mothers were prohibited from eating eggs, meat, fish, or vegetables. In Arusha, adolescent girls were prohibited from eating goat meat similar to Singida where adolescent girls were prohibited from eating chicken thighs. It is believed that by feeding a young child with eggs her hair gets plucked. This study underscores that food taboos and preferences still exist in Tanzania Mainland, and implies gaps in the nutrition education. Thus, nutrition education campaigns and programs should address food taboos and preferences for the meaningful tackling of malnutrition.

www.advancingnutrition.org/sites/default/files/
2023-09/usaid_an_taboos_tfnc_2023.pdf

**Funding:** The U. S. Agency for International Development provided financial support for this article through its flagship multi-sectoral nutrition project, USAID Advancing Nutrition. It was prepared under the terms of contract 7200AA18C00070 awarded to JSI Research & Training Institute, Inc. with a subgrant award to the Tanzania Food and Nutrition Centre (TFNC). The contents are the responsibility of the authors and do not necessarily reflect the views of USAID or the U.S. Government.

**Competing interests:** The authors have declared that no competing interests exist.

## Introduction

Malnutrition has significant health and economic consequences, the most serious of which is death [1–3]. Pregnant and breastfeeding women, adolescent girls, and young children are at high risk of all forms of malnutrition [2,3]. Malnutrition is linked with about 50% of deaths among children below 5 years of age in low- and middle-income countries [1,3]. Of about 50% of deaths, 19% are due to being underweight, 14.5% to stunting, and 14.6% to wasting [1–3]. The recent Tanzania Demographic and Health Survey (TDHS) revealed that 30% of children below 5 years of age are stunted (33% in rural and 21% in urban) and, 3% are wasted [4]. Further, Khan and colleagues documented that, only one-third of children aged 6–23 months receive optimal complementary diet in developing countries [5]. In Tanzania, 8% of children aged 6–23 months were fed a minimum acceptable diet, 19% received the minimum number of food groups, and 33% were fed the minimum number of times [4]. Shreds of evidence have shown that children aged 6–23 months are at high risk of being stunted if they don't receive adequate amount, frequency, and consistency of complementary foods [2,3].

The empirical evidence reported that adolescent girls, pregnant women, and breastfeeding mothers are the best upholders of social balance and because of this it is of great importance that adequate care and attention in the matter of health, nutrition, education, or matters related to their social and economic development [5–8]. However, South Asia and Sub-Saharan Africa are home to over two-thirds of adolescent girls and women who are underweight and, 60% of nutritional anaemia [2,3,7]. Evidence shows that less than one-quarter of adolescent girls and women in Tanzania are underweight but, only 25% consumed diets that meet the minimum dietary diversity [4]. The root causes of malnutrition in pregnant and breastfeeding women, adolescent girls, and young children are complex and multidimensional and call for more evidence to inform the nutrition programs in low- and middle-income countries [1,3]. Most households in sub-Saharan African countries depend on carbohydrate-rich staples while consuming small amounts of nutritious foods such as animal products, fruit, and vegetables [5,6]. Therefore, diets lack sufficient amounts of nutrients needed for good nutrition and health [5,6,9]. On the other hand, insufficient amounts of adequate protein in the diet are further depleted by prohibitions that forbid or discourage eating particular foods and adversely affect the health status of a population [9,10].

Taboos are generally understood as restrictions and prohibitions that are linked to the sacred and are often found to be the basis of maintaining social order. Violations of taboos are abhorred, as it is believed that they incur bad omens and earn social displeasure for the person who violates them. Taboos are associated with bizarre, abominable, and unnatural behavior. They also reflect ideas of social and cultural prohibition. They are linked to almost all aspects of life, and the consequences are not limited to religious spheres but extend to every social and cultural aspect of life. They encompass the practices of do's and don'ts that are still respected today [11,12]. Food preferences encompass the selection of food by individuals, which is influenced by a range of biological and economic considerations such as taste, value, quality, and preparation methods, including cooking fuel and other necessary culinary tools [12]. In this article, we utilize the insights from the UNICEF conceptual framework on maternal and child nutrition [13]. Taboos are considered a macro-environmental "distal factor" that directly and indirectly influences food preferences by shaping diet-related behaviors. These influences may have long-term consequences for the health and nutrition of a population [13]. Understanding food taboos and individual preferences in food items is critical to the development of effective nutritional programmes and educational messages. However, food taboos and preferences have not been fully investigated and their prevalence is not limited to the people of one country, they are found all over the world and some of them continue to be practiced despite the

staggering scientific advances [8–10]. Taboos are deep seated and widespread that they are bound to have an impact on the day-to-day life of many societies [11,12]. Furthermore, perceptive and other cognitive factors come into play and this makes it possible for the emergence of patterns of preferences and these develop readiness to react toward the given situation [8–10]. It is imperative to consider food taboos and preferences from the perspective of existing nutrition challenges facing women, adolescent girls, and young children in sub-Saharan African countries.

The UNICEF's conceptual framework on maternal and child nutrition lists cultural norms, taboos, and preferences among the underlying causes of malnutrition that shape society members' thoughts, ideals, behaviour, and actions [13]. In Malaysia, pregnant women are prohibited from eating fish caught by other people besides the pregnant woman's husband, coconut and cabbage fearing difficulties during delivery, and fused double banana for fearing twin pregnancy [14]. In Ethiopia, pregnant women are forbidden to take all foods that look whitish such as milk products, fatty meat, etc. fearing plaster on the body of the newborn [15]. In Nigeria, pregnant women are prohibited from the consumption of meat, milk, and cheese which are highly needed during pregnancy [16]. In Kenya, pregnant women are restricted from consuming food items such as liver, intestines, kidney, milk, and eggs fearing to cause obstructed labour [17]. A survey conducted in the southern region of Tanzania targeted 954 pregnant women to assess their knowledge of prohibited foods during pregnancy. Findings revealed that 372 respondents identified fish, 314 identified various types of meat (including farm animals and bush meat), 68 identified leftovers, and 34 identified eggs as taboo during pregnancy [18] and, a qualitative research study conducted in northern Tanzania, researchers explored the perspectives and daily dietary practices of Maasai pregnant women. The study findings emphasized the traditional advice given to pregnant Maasai women, which includes avoiding unpasteurized milk, meat, milk from cattle other than their own, eggs, sweet foods, and butter to prevent the birth of 'big' babies and complications during childbirth. [19].

Food taboos play a significant role and may act as barriers or facilitators of food preferences. Embedding a cultural understanding is important as global initiatives in developing effective nutrition promotion strategies [13]. Tanzania is implementing a five-year second National Multisectoral Nutrition Action Plan (NMNAP-II) from 2021/22 to 2025/26 with specific targets aimed at improving women, young children, and adolescent girls' maternal health and nutrition status [20], however, it is quite challenging to implement the programs with the dearth of information pertaining the food taboos and preferences. The present study aimed to explore the food taboos and preferences among adolescent girls (aged 15–19 years), pregnant women, breastfeeding women, and young children (aged 6–23 months) in Tanzania's Mainland. An in-depth understanding of food taboos and preferences will contribute to addressing the malnutrition situation in Tanzania and shaping diet-related behaviours among adolescent girls, pregnant women, breastfeeding women, and young children aged 6–23 months. Further, the information will inform targeted educational interventions aimed at improving dietary practices and nutritional outcomes in Tanzania.

## Methods

### Study design

A qualitative study employing ethnography was conducted to explore information from Key informant interviews and Focus Group Discussions (FGDs). Semi-structured interview guiding questions were used to capture the insights and understandings of food taboos of society members at different levels and positions, adolescent girls (aged 15–19 years), pregnant women, breastfeeding women and their children (aged 6–23 months) and identify

opportunities and gaps for nutrition-sensitive programs. The ethnographic method was used because we were interested in exploring the context, research participants' views, and complex interactions **S1 Checklist**.

## Study settings

The study was conducted in one rural and one urban ward in the seven regions of Tanzania's Mainland: Dar es Salaam; Kigoma; Arusha; Lindi; Mbeya; Singida and; Mwanza. Participant recruitment was carried out from February 23rd, 2022 to February 22nd, 2023.

## Study population

The parent study included: Pregnant women; breastfeeding mothers and their children aged 6–23 months and; adolescent girls aged 15–19 years. For triangulation purposes, we included a limited number of key informants in the study to gain an in-depth understanding of the factors influencing food taboos including district nutrition officers, community leaders, and representatives of civil society organisations working in nutrition programs in their respective study areas.

## Selection of study participants

We used a multistage sampling technique to select one region from each zone. In the second stage, each regional study team purposefully selected one rural and one urban ward. The third stage involved the selection of study participants guided by the recommendations of the simulation and guidelines in qualitative research, for focus group discussions (FGD) and in-depth interviews (IDI) [21–23].

## Data collection

**Focus Group Discussions *(FGDs)*.**   These were conducted with adolescent girls (aged 15–19 years), pregnant women, breastfeeding women, and proxies to their children (aged 6–23 months). In each FGD, the total number of participants ranged from 6 to 12. FGD participants included people with homogeneous socio-demographic characteristics. In recruiting study participants, we observed the rural and urban nature of people's behaviours to gauge perspectives on food taboos and preferences. Tables 1 and 2 summarise the study FGD sample. Community leaders assisted with the identification of the FGD participants in their respective communities. The FGD interview guide is annexed as the **S1 Text**.

*In-depth interviews (IDI)*. Were conducted with adolescent girls (aged 15–19 years), pregnant women, and breastfeeding women, and their children (aged 6–23 months), district nutrition officers, community health workers, significant others, influential persons, community leaders (religious leaders, and local government/village leaders), and representatives of civil society organisations working in nutrition programs in their respective study areas. The interviewing process continued until a point was reached whereby no new information emerged-the data saturation point. We carefully considered the situation and determined that additional sampling was unnecessary. The IDI interview guide is annexed as the **S2 Text**.

**Data analysis.**   Verbatim transcription of the FGD and IDI sessions was done by research assistants or note-takers. The qualitative personnel verified all transcriptions to ensure that transcription was accurate by listening to the audio and making sure what was written was similar to what was in the audio [21–24]. The interviews were conducted in the Swahili language and simultaneous transcription and translation were used [21–24].

The transcribed verbatim were coded thematically for content analysis and interpretation. We coded our transcripts and analysed them using MAXQDA software. The team of

**Table 1. FGDs study participants in the seven regions of Tanzania Mainland.**

| Zone | Region | District | Respondent group | | | FGDs |
|---|---|---|---|---|---|---|
| | | | Pregnant women | Breastfeeding mothers | Adolescent girls | |
| Southern highlands zone | Mbeya | Mbeya city council -urban | 1 (7 participants) | - | 1 (12 participants) | 2 |
| | | Busokelo district council-rural | 1 (9 participants) | 1 (12 participants) | - | 2 |
| Eastern zone | Dar es Salaam | Kinondoni municipal council-urban | 1 (9 participants) | - | 1 (8 participants) | 2 |
| | | Ubungo-rural | - | 1 (10 participants) | - | 1 |
| Lake zone | Mwanza | Nyamagana-urban | 1 (9 participants) | - | | 1 |
| | | Sengerema-rural | - | 1 (9 participants) | 1 (8 participants) | 2 |
| Western zone | Kigoma | Kigoma- urban | - | 1 (9 participants) | 1 (8 participants) | 2 |
| | | Kakonko- rural | 1 (8 participants) | - | 1 (12 participants) | 2 |
| Southern zone | Lindi | Lindi- urban | 1 (7 participants) | - | - | 1 |
| | | Liwale- rural | - | 1 (8 participants) | 1 (11 participants) | 2 |
| Northern zone | Arusha | Arusha- urban | 1 (12 participants) | - | 1 (9 participants) | 2 |
| | | Longido- rural | 1 (6 participants) | 1 (10 participants) | - | 2 |
| Central zone | Singida | Singida- urban | - | 1 (8 participants) | 1 (8 participants) | 2 |
| | | Mkalama -rural | 1 (8 participants) | - | 1 (8 participants) | 2 |
| | **Total FGDs per group** | | **9** | **7** | **9** | **25** |

researchers under the leadership of persons with expertise in qualitative methods read the text several times before developing codes [21]. The second person (also trained and experienced) coded a subset of transcripts using the developed codebook to assess the quality of the data and their reliability for interpretation and final use [21,22]. The codebook structure was developed iteratively after the completion of transcription. Themes were discussed and agreed upon by all researchers. The thematic analysis was carried out by assigning data into relevant codes to generate categories based on study objectives [21–24].

**Ethical considerations.** This study obtained ethical permit from the National Health Research Ethics Review Committee (NatHREC) at National Institute for Medical Research with reference no. NIMR/HQ/R.8a/Vol.IX/3930. We also obtained permission to conduct a study at regional, district and ward levels in all study regions. All participants provided written consent prior to participating in the study and did not receive financial compensation for their participation. Furthermore, written informed consent was obtained from the parent/guardian of each participant under 18 years of age. The ethics training was given to the research team for empowering them to: recognize power dynamics and its potential influence; and reflect on personal biases and assumptions during the process of data collection. Anonymity was maintained throughout the study and data sets will be destroyed when the project is concluded.

**Table 2. FGDs study participants from rural and urban settings of Tanzania Mainland.**

| Place | Respondent | | | Total |
|---|---|---|---|---|
| | Pregnant women | Breastfeeding mothers | Adolescent girls | |
| Urban | 5 (44 participants) | 2 (17 participants) | 5 (49 participants) | 12 (110 participants) |
| Rural | 4 (31 participants) | 5 (49 participants) | 4 (39 participants) | 13 (119 participants) |
| **Total** | **9 (75 participants)** | **7 (66 participants)** | **9 (88 participants)** | **25 (229 participants)** |

# Results

## Food taboos

Food taboos and restrictions are common in most of the regions that have been studied. There are some similarities and variations across the regions and population groups studied. The taboos and restrictions for pregnant women are mostly aimed at protecting the health of the mother and offspring. This includes prohibiting the consumption of animal products such as eggs as a source of protein.

**Pregnant women.** In our current study, we have observed that different regions and communities have their food taboos for pregnant women, as outlined in Table 3. It is crucial for the health of both the unborn baby and the mother that pregnant women consume an adequate amount of food containing essential nutrients from various food groups. Food taboos that restrict food consumption during pregnancy can lead to low nutrient intake and, ultimately, malnutrition, which can have negative effects on the health of the unborn baby and the mother. In the Arusha region, specifically in the Maasai community, there is food taboos related to the amount of food that pregnant women should consume. In this community, it is prohibited for pregnant women to have large meals due to fears of giving birth to a large baby, experiencing pain from perineal tears, and needing a cesarean section. Usually, other women and the spouses of pregnant women are responsible for ensuring that these food taboos are followed. One of the respondents pointed out that.

*"Pregnant women are told they shouldn't eat large meal[s] as it might result in giving birth to a big baby. . . . . . . and they are encouraged to drink porridge" (KII, DNuO, Longido DC, rural).*

**Table 3. Restricted food items among pregnant women in Tanzania Mainland.**

| Taboo | Reasons | Region/Place |
|---|---|---|
| Lemon | Dry blood<br>Feeling dizzy<br>Causes blood thinning | Dar es Salaam, Ubungo MC—rural, Arusha CC—urban |
| Peppers | Baby will be born with red eyes | Arusha, Arusha CC—urban |
| Pineapple | A baby will be born with skin patches | Mbeya, Busokelo DC—rural |
| Pork | Religious reasons | All |
| Eggs, chicken | Giving birth to a baby with no hair<br>Bad for the child<br>Giving birth to baby with a swollen head<br>Delivering an unhealthy infant | Arusha, Arusha CC—urban, Longido DC—rural<br>Mbeya, Busokelo DC—rural,<br>Dar es Salaam, Ubungo MC—rural<br>Singida, Mkalama DC—rural<br>Mwanza, Nyamagana MC—urban |
| Clay/soil/charcoal | Has worms leading to anaemia<br>Has bacteria<br>Higher risk of giving birth to an ill baby<br>When a child is born, he or she will be eating a lot of soil as well<br>Causes constipation and hernia<br>Causes appendicitis<br>Affects the organism inside [baby] | Singida Singida, MC—urban<br>Mbeya, Mbeya CC—urban<br>Mwanza, Nyamagana MC—urban, Sengerema DC—rural |
| Eating until one is full/large meal | Results into a big baby hence surgery during delivery | Arusha, Longido DC—rural |
| Kambale (fish) | Delivering a child with a physical disability | Mwanza, Nyamagana MC—urban |
| Kuhe (fish) | When she starts labour, it will be on and off | Kigoma, Kigoma MC—urban |

In the Mbeya region, pregnant women are forbidden from eating eggs and certain fish varieties such as catfish and marbled lungfish (referred to as Kambale and Kamongo in Swahili). It is believed that consuming these could lead to babies being born without hair, with swollen heads, or with physical disabilities:.

*". . . Women should not eat eggs because they can give birth to a big baby whereby delivery by operation will be needed" (IDI, significant other, Ubungo, rural).*

*". . . if a woman has not given birth, she is not allowed to eat eggs and if she eats [she] can give birth to a child with no hair" (KII, influential person, Busokelo DC, rural).*

We have also found that some communities believe that consuming lemon during pregnancy can lead to negative health outcomes, including giving birth to low birth weight babies. Nonetheless, citrus fruits are abundant in vitamin C, essential for the well-being of unborn babies and expectant mothers. One of the participants in the study made a pertinent observation.

*"We are only advised by the community around us. . . if you eat lemons when you have low blood, it may cause you problems. If you eat lemons, you will start to feel dizzy. . . also with pepper, you will give birth to a child with red eyes; the child will suffer from an eye disease" (FGD, pregnant women, Arusha CC, urban).*

In urban areas, pregnant women are not allowed to consume food leftovers (known as "Kiporo" in Swahili) due to the belief that it may cause defecation during delivery. The reasons given for this prohibition reflect misconceptions that may limit nutrient intake, as consuming leftover food is a common practice. However, restricting the consumption of leftover food may help lower the risk of transmitting foodborne diseases and associated problems.

*". . . Leftover food [Kiporo in the Swahili language] has lost nutrients and if a pregnant woman consumes, she will get illness" (FGD, pregnant women, Kinondoni MC, urban).*

*"In our society, we don't allow pregnant women to eat leftovers. . . . . . . . . if eaten she will defecate more during childbirth" (FGD, pregnant women, Kinondoni MC, urban).*

**Breastfeeding mothers.** Respondents noted variations in food taboos for breastfeeding mothers across study areas of Singida MC, Mkalama DC, Arusha CC, Longido DC, and Ubungo MC. All taboos related to ensuring the well-being of a mother and newborn baby typically last for a few weeks after childbirth as shown in Table 4.

It is taboo for breastfeeding mothers to eat hard (e.g., ugali, rice) and cold foods. Instead, postpartum mothers should eat warm, soft, and watery foods such as nutritious soft porridge composed of a variety of food groups (rice, maize, sorghum, millet, peanut, or Blue Band and milk); *Uji wa lishe* (soft porridge made of a mixture of cereals with peanut butter or margarine and milk); meat soup; or cooked stewed banana in the days immediately following delivery. Eating chilled or hard foods, such as stiff porridge made up of millet, sorghum, beans, grounded local green vegetables (referred to as ndago in Swahili), or sardines and rice is prohibited during the first few days after delivery. This food taboo relates to preventing constipation, stomach aches, and difficulty in bowel movements, assisting in recovery from perineal

**Table 4. Restricted food items among breastfeeding mothers in Tanzania Mainland.**

| Taboo | Reasons | Region/Place |
|---|---|---|
| Hard foods | Body has not recovered yet so difficult to digest hard foods (e.g., stiff porridge made of millet, sorghum) Having ulcers in the stomach Do not stimulate breast milk production | Singida, Singida MC—urban Arusha, Arusha CC—urban Dar es Salaam, Kinondoni MC—urban |
| Ugali | Difficulties in bowel movement The stomach is still not in a situation where it can receive such a type of food | Arusha, Arusha CC—urban Singida Dar es Salaam, Kinondoni MC—urban |
| Beans | Stomach ache and constipation | Singida, Mkalama DC—rural |
| Nut grass | Stomach ache and constipation | Singida, Mkalama DC—rural |
| Sardines | Stomach ache and constipation | Singida, Mkalama DC—rural |
| Rice | Difficulties in bowel movement Constipation The stomach is still not in a situation where it can receive such a type of food | Dar es Salaam, Kinondoni MC—urban, Ubungo DC—rural |

tears after giving birth, as well as promoting adequate milk production. Some reasons given for this were as follows:

*"After childbirth, lactating women should eat food that stimulates milk production including meat soup, porridge, mashed cooked banana, not ugali based on locally available food" (CSO, Arusha, MC, urban).*

*"She is supposed to eat those soft foods such as cooked bananas, soft porridge, or soft cooked rice mixed with ground groundnuts and oil until [she] recovers [enough] to eat hard food especially if she has perineal repair following [a] tear following childbirth trauma" (IDI, significant other, Singida MC, urban).*

**Young children of 6–23 months.** We asked participants about food taboos for their children aged 6–23 months, some reported that feeding eggs to young children is a taboo.

*"Maybe there are customs like food taboos. Others say if a child is fed with eggs his/her hair gets plucked, but he/she doesn't know that an egg helps a child" (FGD, adolescent girls, Singida MC, urban).*

Participants also said that the community believes that some staple foods lack nutrients (e.g., ugali with mlenda) or are hard for a baby to chew and swallow (ugali, rice, cassava). As such, these foods are taboo for young children.

*"Ugali and Mlenda, it has no nutrients. Rice, the child will not be satisfied" (FGD, adolescent girls, Mkalama DC, rural).*

Surprisingly, most of the foods excluded from children's diets are types of staples that are commonly used, affordable foods and can be nutritious for a baby and promote healthy growth and development, such as ugali and rice. The finding implies there is a need for community awareness to address misconceptions (unhealthy food practices) to ensure children are fed a balanced diet using locally available foods. It is important to note that these food restrictions are associated with mothers' preference for their babies to be healthy.

**Adolescent girls.**   Food taboos among adolescent girls vary across communities; adolescent girls in the regions of Singida, Dar es Salaam, and Kigoma are allowed to eat most foods as compared to other regions. Most restricted foods are good sources of protein, which plays an important role in the normal functioning of the body as highlighted in Table 5. In Arusha, urban respondents attested that:

> *"We used to ask why we are not allowed to eat marbled lungfish; they told us a mother or the baby girl is not allowed to eat marbled lungfish because she will develop additional breasts"* (FGD, adolescent girls, Arusha CC, urban).

In urban Mbeya and rural Singida regions, adolescent girls are prohibited from consuming animal meat, such as goat meat and chicken thighs, respectively.

> *"There are foods that are culturally not allowed to be eaten by female. . ... Our tradition does not allow a female to eat goat meat"* (FGD, adolescent girls, Arusha CC, urban).

> *". . ... if a chicken is slaughtered in the family, parents are given a priority to eat meat like a thigh"* (FGD, adolescent girls, Mkalama DC, rural).

## Food preference

We observed both healthy and unhealthy food preferences exist among various study communities. Various food preferences vary for pregnant women, breastfeeding mothers, and their children across regions. The factors influencing people's food choices included the availability and affordability of food, taste preferences, cultural identity, socialization, and urbanization.

**Pregnant women.**   The food preferences among pregnant women were similar in both rural and urban areas:

> *"Vegetables such as pumpkin leaves, cabbage, 'figiri' or Chinese as well as cooked fish and meat and eggs are the preferred food"* (FGD, pregnant women, Busokelo DC, rural).

Despite the importance of vegetables, few women mentioned a preference for vegetables and this can limit intake of nutrients such as vitamins, minerals and fibers, important for the health of a mother and baby.

**Table 5. Restricted food items and reason behind among adolescent girls in Tanzania Mainland.**

| Food Conditions/Feeding Behaviour | Reasons | Region/Place |
|---|---|---|
| Chicken thighs | Reserved/prioritised for parents | Singida, Mkalama DC—rural |
| Pork | Religious reasons | All regions |
| Marbled lungfish | Woman will develop additional breasts | Arusha, Arusha CC—urban |
| Meat from the goat | Reserved for men | Mbeya, Mbeya CC—urban |
| Uncooked rice | Vaginal candidiasis—whitish water will come out of the from vagina, f form fungus | Dar es Salaam, Ubungo DC—rural |

*"Consumption of vegetables is very low, which is very important in building the body and immunity for body protection and healthy" (IDI, CSO, Arusha CC, urban).*

In the urban areas the socio-economic status of the family dictated pregnant women's food preferences, since they eat what they can afford:

*"You cannot force yourself to eat something good if you can't afford to buy, so you have to eat what is available" (FGD, pregnant women, Arusha CC, urban).*

**Breastfeeding mothers.** Postpartum and lactating mothers prefer foods that are quick to prepare, in season, affordable, tasty, unique, and provide excitement and happiness when consumed. *"These are the food[s] with good taste. For instance, the taste is different for one who consumes rice served with beans compared to one [who] consumes stiff porridge served with beans. You find most of the women enjoy the taste when eating these foods" (IDI, VEO, Ubungo MC, rural).*

**Young children of 6–23 months.** Participants reported various appropriate food for their children of 6–23 months and reasons for feeding them. The following foods were reported; porridge made of maize, cassava, or millet flour (cereals) and sometimes mixed with groundnuts or milk (as source of protein), before adding bananas, cassava, and sweet potatoes (carbohydrates). One participant attested that—

*". . . we have been learning that the children should take different types of food rich in proteins (fish, beans); carbohydrates (stiff porridge, sweet potatoes, rice); vitamins and minerals (watermelon, Chinese, pumpkin leaves)" (IDI, CHW, Kakonko DC, rural).*

**Adolescent girls.** The exception is the Mwanza and Kigoma regions, where girls preferred ugali. Most adolescent girls preferred to consume energy-dense foods that can predispose them into malnutrition-related health problems as shown in Table 6.

*"[I prefer eating] French fries served with chicken, eggs, barbeque meat, fish, sausage, or plain chips" (FGD, adolescent girls, Kinondoni MC, urban).*

Adolescents have a limited understanding of appropriate eating practices. Peer pressure, ready availability, low cost, good taste, unique food, and ease of carrying usually drive their preferences.

*"In my opinion, they go for easily accessible food in terms of cost, because she may be a college or high school student struggling to get other needs. So, she will go to cheap food regardless of the taste" (IDI, CSO, Arusha CC, urban).*

*"Ahh social media influences adolescent life, you find someone is interested to live like Paula (a local celebrity) including copying food she eat[s]" (FGD, adolescent girls, Arusha CC).*

The findings indicate a missed opportunity to mitigate unhealthy eating habits, dietary misconceptions, and their impact on adolescent girls.

**Table 6. Food items preferred by adolescent girls in Tanzania Mainland.**

| Preferred food item | Reason | Region/Place |
|---|---|---|
| Fried potatoes | Easy to carry from one place to another, and easy to prepare<br>Simple food<br>Easily available | Kigoma, Kigoma MC—urban<br>Dar es Salaam, Kinondoni MC—urban<br>Arusha, Arusha CC—urban, Longido DC—rural<br>Singida, Singida MC—urban<br>Mwanza, Sengerema DC—rural |
| Rice | High appetite for rice | Kigoma, Kigoma MC—urban Mwanza, Sengerema DC—rural<br>Nyamagana MC—urban |
| Fried cassava | Stays in the stomach for longer<br>Good taste<br>Simple food | Kigoma, Kigoma MC—urban<br>Dar es Salaam, Kinondoni MC—urban<br>Arusha, Arusha CC—urban |
| Fish | Nutritious, tastes good, availability | Mwanza, Nyamagana MC—urban<br>Mbeya, Mbeya CC—urban |
| Ugali | Availability, staple food | Mwanza, Sengerema DC—rural<br>Singida, Singida MC—urban<br>Mbeya, Mbeya CC—urban |
| French fries with fried eggs or chicken | Uniqueness/special, tastes good<br>Simple food, easily available | All regions |
| Ice cream, cakes, chocolate, pancakes | Tastes good | Dar es Salaam, Kinondoni MC—urban<br>Arusha, Arusha CC—urban |

## Discussion

This qualitative study explored food taboos and preferences of different groups and regions. The information presented here is based solely on the participants' responses and is specific to the areas covered by the study. Although we aimed to gather information on common food taboos and preferences of pregnant women, breastfeeding mothers, adolescents, and young children in each region, there may be variations that we did not capture. So, it is important to keep in mind that the findings of this study should be understood as applicable only to those areas and not representative of the taboos and preferences of Tanzanian women as a whole. Further, excluding men and elderly women in this study misses the opportunity to deeply understand the subtleties of food taboos and their cultural underpinnings. Nonetheless, this study provides valuable insights into the food habits of different groups and regions, which can help inform policies and interventions for improving nutrition and public health in Tanzania.

Food taboos are common practice in many developing countries, regardless of whether they are located in urban or rural areas. Several studies, including those conducted by Chakona and Shackleton [25], and Kavle and Landry [26], have shed light on this phenomenon. For example, research conducted in rural Tajikistan has shown that food taboos during pregnancy persist because they are believed to protect and support maternal health [12]. However, most of these food taboos are not related to nutrition or health promotion [12,24]. On the other hand, women from the Maasai community in Tanzania recognize the importance of proper nutrition during the prenatal period, as it is crucial for both their health and the health of their babies [16,27]. Our study has revealed some interesting variations in the types of food restrictions and the reasoning behind them in different study areas.

The present study findings indicate that food taboos and restrictions related to unhealthy food, high protein food (e.g. eggs, chicken, and fish), as well as vitamin C (found in citrus fruits and pepper) and mineral-rich food (such as meat, chicken, and fish) have been reported in

various countries, including Ghana by Arzoaquoi et al. [10], Nigeria by Ugwa [16], and Kwa Zulu-Natal, South Africa by Ramulondi et al. [28]. These similarities imply the existence of interrelated continental food taboos and restrictions, particularly in low- and middle-income countries (LMICs).

It has been observed that even though mothers try to eat healthy during pregnancy, they tend to consume unhealthy food after giving birth and during lactation. This is consistent with the findings of studies conducted by Jardí et al. [29] and Poulain et al. [30], which also reported a decrease in the consumption of healthy food and an increase in the consumption of junk food from pregnancy to the postpartum period. This could be because, during pregnancy, mothers are more concerned about the negative health effects of unhealthy food and receive more nutritional education during antenatal care (ANC) which emphasizes the importance of nutrition. Moreover, healthy eating habits during pregnancy may be influenced by certain food taboos and norms that are no longer applicable after childbirth. Therefore, it is important for stakeholders in nutrition and reproductive, maternal, newborn, child, and adolescent health (RMNCAH) to investigate the reasons behind the decrease in healthy food consumption after childbirth and adapt accordingly on time.

Within our study population, it has been observed that there are no taboos or restrictions on the consumption of unhealthy foods by children below two years of age. This indicates a liberal approach towards food consumption, allowing children to consume any food. It is noteworthy that the Tanzania Demographic and Health Survey Program 2022 report has reported a mere 9 percent of unhealthy food consumption in this age group, reflecting the successful implementation of healthy food consumption practices for children under two [3]. However, caregivers and mothers have expressed their concerns regarding the use of ready-made foods for children, citing the presence of chemicals and expiration dates/storage as issues of concern. It is imperative to address these concerns and ensure the safety and well-being of young children.

The present research demonstrates that lactating mothers tend to restrict the consumption of staple foods, such as rice, cassava, and ugali, for their children due to concerns about the risk of choking. However, such constraints may have negative implications for child nutrition, particularly during the complementary feeding period, which commences after six months of exclusive breastfeeding. Failure to provide children over six months of age with a healthy and diverse diet may result in malnutrition [16,28]. Therefore, it is imperative to broaden the age group categorization of sociocultural factors that influence food consumption of age-specific taboos and restrictions and their impact on nutritional outcomes..

Most existing taboos and norms for postpartum and lactating mothers relate to biology and health promotion, although, the reason given for adherence sometimes does not make the nutrition benefit clear. For instance, promoting the consumption of hot/warm liquid foods believed to stimulate breast milk production also reduces the risk of foodborne diseases and, provides mothers with enough body fluid to promote milk production. This study's results provide compelling evidence that lactating mothers should be cautious when consuming cold foods, aligning with a similar study conducted in West Bengal, India [31]. There are few studies on food taboos conducted in Tanzania, which highlights the relevance of the current study when making dietary decisions for optimal lactation outcomes. In addition, advice to consume special or high-protein food plays a vital role in the recovery and production of breast milk and advice to consume soft food reduces the risk for constipation. Health education interventions should focus on giving women accurate nutrition information to counteract unhealthy taboos.

Despite the national effort to promote good nutrition for adolescent girls, unhealthy food consumption habits and preferences such as the consumption of fast food in high amounts prevail. This is not just a problem in Tanzania, but a global trend that has become a cause for concern. Research shows that in recent years, the consumption of processed foods like bread,

biscuits, popcorn, processed cereals, meat, potatoes/cassava chips, and margarine has increased in Tanzania due to urbanization, income growth, and changing lifestyles [32]. This shift towards processed foods is also seen in other low- and middle-income countries, where adolescent girls in urban areas are more likely to consume energy-dense and high-fat foods [33]. However, our study found that the preference for processed foods is evident in urban and rural areas of Tanzania. This suggests increasing interaction and diffusion of ideas and life-styles between rural and urban areas in modern Tanzania. It's important to note that there are constant interactions between rural and urban communities, so it's not surprising that food preferences are similar among participants from both settings.

Previous studies found a strong association between food availability, affordability, desir-ability, and taste with unhealthy eating habits. Urbanization, socialization, identity, and mis-conceptions shaped the food preferences of adolescent girls, while a sense of excitement and happiness influenced the choices of postpartum and lactating women. Furthermore, adherence to food taboos during pregnancy was crucial in protecting the health of pregnant women and their babies, with reasons for following taboos similar to those found in West Bengal, Senegal, and South Africa [25,31]. These findings suggest that healthy taboos can play a significant role in maintaining the well-being of expectant mothers and their infants.

Misinformation about nutrition and food is becoming a serious issue, especially among adolescent girls in low- and middle-income countries. This is because there are many gaps in their knowledge about nutrition. To address this problem, we need to focus on correcting mis-conceptions about healthy food and emphasizing the importance of consuming it in its entirety. This will help to improve the nutritional status of both women and children. For example, consuming pepper and lemon during pregnancy has been associated with red eyes in newborns and anaemia in mothers. Similarly, consuming uncooked rice has been linked to vaginal candidiasis in adolescent girls. These findings highlight a significant knowledge gap among communities in understanding early signs of health problems. It is important to address these misconceptions and educate people on managing their health problems on time, rather than avoiding certain types of food.

In our study, fibers and fruits were not preferred, only postpartum and lactating mothers reported the consumption of fruits. These findings are similar to other studies that show low intake of fruits and vegetables among pregnant women, lactating women, adolescent girls, and children especially in low- and middle-income countries [34,35]. Respondents only reported taboos related to excluding citrus fruits (lemons) [36]. Based on the variety of fruits and vegeta-bles available, findings may signify economic constraints rather than behavioural factors to vegetables and fruits.

## Conclusions

This study underscores that food taboos among adolescent girls (aged 15–19 years), pregnant women, breastfeeding women, and children (aged 6–23 months) still exist in Tanzania Main-land and imply gaps in the nutrition programs. There is a need to strengthen nutrition cam-paigns and programs that should address food taboos and preferences for the meaningful tackling of malnutrition in Tanzania's Mainland. Further research with a longitudinal design are recommended to analyze the distal drivers of malnutrition and specifically explore whether the food taboos are hampering nutrition outcomes among adolescent girls, pregnant women, breastfeeding women, and young children.

## Supporting information

**S1 Checklist. STROBE checklist.**
(DOCX)

**S1 Text. FGDs interview guide.**
(DOCX)

**S2 Text. IDI guide.**
(DOCX)

## Author Contributions

**Conceptualization:** Aika Lekey, Ray M. Masumo, Zahara Daudi, Winfrida Onesmo, Germana H. Leyna.

**Formal analysis:** Aika Lekey, Ray M. Masumo, Theresia Jumbe, Mangi Ezekiel, Zahara Daudi, Nangida J. Mchome, Glory David, Winfrida Onesmo, Germana H. Leyna.

**Investigation:** Aika Lekey, Theresia Jumbe, Mangi Ezekiel, Winfrida Onesmo.

**Methodology:** Aika Lekey, Ray M. Masumo, Mangi Ezekiel, Zahara Daudi.

**Project administration:** Germana H. Leyna.

**Supervision:** Ray M. Masumo, Theresia Jumbe, Germana H. Leyna.

**Writing – original draft:** Aika Lekey, Ray M. Masumo, Theresia Jumbe, Mangi Ezekiel, Zahara Daudi, Nangida J. Mchome, Glory David, Winfrida Onesmo, Germana H. Leyna.

**Writing – review & editing:** Aika Lekey, Ray M. Masumo, Theresia Jumbe, Mangi Ezekiel, Zahara Daudi, Nangida J. Mchome, Glory David, Winfrida Onesmo, Germana H. Leyna.

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
