## [Decision Letter · Decision Letter 0]

7 Jun 2024

PGPH-D-24-00652

Food taboos and preferences among adolescent girls, pregnant women, breastfeeding mothers and children of 6-23 months in Mainland Tanzania

Dear Dr. Masumo,

Thank you for submitting your manuscript to PLOS Global Public Health. After careful consideration, we feel that it has merit but does not fully meet PLOS Global Public Health’s publication criteria as it currently stands. Therefore, we invite you to submit a revised version of the manuscript that addresses the points raised during the review process.

We look forward to receiving your revised manuscript.

Kind regards,

Hasanain Faisal Ghazi, phd

Academic Editor

Journal Requirements:

Additional Editor Comments (if provided):

please respond to reviewers' comments

Reviewers' comments:

Reviewer's Responses to Questions

**Comments to the Author**

1. Does this manuscript meet PLOS Global Public Health’s publication criteria? Is the manuscript technically sound, and do the data support the conclusions? The manuscript must describe methodologically and ethically rigorous research with conclusions that are appropriately drawn based on the data presented.

Reviewer #1: Yes

Reviewer #2: Yes

2. Has the statistical analysis been performed appropriately and rigorously?

Reviewer #1: N/A

Reviewer #2: Yes

3. Have the authors made all data underlying the findings in their manuscript fully available (please refer to the Data Availability Statement at the start of the manuscript PDF file)?

Reviewer #1: Yes

Reviewer #2: Yes

4. Is the manuscript presented in an intelligible fashion and written in standard English?

Reviewer #1: Yes

Reviewer #2: Yes

5. Review Comments to the Author

Reviewer #1: Food taboos and preferences among adolescent girls, pregnant women, breastfeeding mothers and children of 6-23 months in Mainland Tanzania

Title:

The title is descriptive and clearly outlines the focus and scope of the study. However, it could be improved for clarity. Here is a revised version: " Food taboos and preferences among adolescent girls, pregnant women, breastfeeding mothers, and children aged 6-23 months in Mainland Tanzania" I added "aged" before "6-23 months" to make it clearer that the authors are referring to the age range of the children. Additionally, I added a comma after "breastfeeding mothers" to maintain parallel structure with the rest of the list. This may be considered.

Introduction:

The introduction provides insight into the study's context and rationale. However, to enhance clarity, authors may consider incorporating a paragraph elucidating the theoretical definition of food taboos, food restrictions, and food preferences. Furthermore, augmenting the discussion with relevant literature on how these taboos and restrictions, and preferences shape diet related behaviours that may have long-term consequences for the health and nutrition of a population.

The concluding sentences in the introduction (The findings of this study will provide information to help improve vital maternal and adolescent nutrition interventions in Tanzania. The insights gained to support the prevention and management of micronutrient deficiencies and integrated management of acute malnutrition by providing education on food taboos and preferences that may affect maternal and adolescent nutrition) are very generic and vague. It is essential for the authors to contextualize the significance of understanding prevalent food taboos/restrictions/preferences within the specific scope of this research. By explicitly stating how this understanding of food taboos/restrictions will contribute in addressing the malnutrition situation in Tanzania. This could involve highlighting how insights into food taboos/preferences among adolescent girls, pregnant women, breastfeeding women, and young children aged 6-23 months will inform targeted educational interventions aimed at improving dietary practices and nutritional outcomes.

Methods

Data analysis: Please also mention whether you utilised any software for data analysis, such as ATLAS.ti, NVivo etc. (if any).

Ethical consideration: Please mention the ethics training (if any) given to the research team for empowering them to: recognize power dynamics and its potential influence; and reflect on personal biases and assumptions during the process of data collection.

Results

The results shed light on the dietary preferences/restrictions of the demographic groups which are under study. The data presented in the paper is more about the prohibition/restriction of certain foods during pregnancy and which is explained as being aimed at protecting the health of both the mother and the baby. There is only limited information on food taboos which are based on misconceptions or misinformation, such as the belief that consuming eggs can result in a child being born without hair.

Most of the dietary restrictions reported by pregnant women are align with healthy practices, serving as protective measures against potential health complications. The question arises is that whether these dietary practices (nutrition information), encompassing avoidance of alcohol, cigarettes, soft drinks, and leftover food during pregnancy, can be classified as food taboos? If the answer is yes, how this will help in developing educational interventions aimed at improving optimal dietary practices and nutritional outcomes.

The section ‘gap in nutrition awareness’ is seems like a discussion and appears not supported by the data from this research. The statements such as ‘adolescents have limited understanding on appropriate eating practices and peer pressure usually drives their food preferences’ needs to be supplemented by verbatim.

The results section requires a more robust presentation of data, particularly emphasizing taboos rooted in misconceptions that significantly impact nutrition and health outcomes. A persuasive restructuring is necessary to ensure a coherent flow, supported by empirical evidence within each domain of investigation.

Discussion

In the discussion, the authors note that ‘fibres and fruits were not preferred, only postpartum and lactating mothers reported the consumption of fruits. These findings are similar to other studies that show low intake of fruits and vegetables among pregnant women, lactating women, adolescent girls, and children especially in low- and middle-income countries.' While the current research aims to explore food taboos, preferences, and restrictions, the consumption of fruits, vegetables, milk, etc. is largely influenced by individuals' purchasing power to incorporating them into their regular diet. This low intake is primarily due to economic constraints rather than behavioural factors.

Limitation: The study focused on specific demographic groups (adolescent girls, pregnant women, breastfeeding mothers, and children aged 6-23 months), potentially overlooking other pertinent stakeholders like adult men/women or elderly individuals, who also play roles in shaping food practices within households. Including these people in the research could have provided deeper insights into the subtleties of food taboos and their cultural underpinnings. Understanding the belief systems and misconceptions surrounding these taboos is essential for designing health programs that effectively address them through targeted health messaging.

Reviewer #2: This qualitative study used the ethnography method to explore food taboos and preferences in food items among breastfeeding mothers, pregnant women, adolescent girls, and their young children aged 6-23 months old in Tanzania.

Title

1. It is highly recommended to indicate the nature and topic of the study Identifying the study as qualitative or indicating the approach of ethnography in the title.

Introduction

1. “In Tanzania, a study conducted in the southern part of the country shows about two thirds of the women avoided fish and farm meats [18] and, in the northern part of the country, pregnant women are restricted from consuming unpasteurized milk, meat, or milk from cattle (other than their own), eggs, sweet foods, and butter for preventing complications during delivery, especially associated with the delivery of a large baby [19]. “

Please check if the reference [18] reported this result. In this study, the cross-sectional descriptive study design can only show the proportion of dietary intake of food, but not if a women avoid. The authors should be careful when describing findings of previous studies throughout the text.

In addition to the previous study cited as reference [19], what is already known from previous studies? Please provide a brief summary of existing evidence from literature and clarify the literature gap that the authors addressed in this study.

2. Are there any relevant theory regarding food taboo and preference in Tanzania or other similar cultures? This could help to specify or state research questions.

Methods

1. Provide guiding theory if appropriate, and rationale for the ethnography method.

2. Provide researchers’ characteristics that may influence the research such as

personal attributes, qualifications/experience, relationship with participants, or assumptions; potential or actual interaction between researchers’ characteristics and the research questions, approach, methods, results, and/or transferability.

3. provide information on criteria for deciding when no further sampling was necessary

(e.g., sampling saturation).

4. Did the authors use other techniques to enhance trustworthiness and credibility of data analysis, such as audit trail, in addition to member check? If yes, please provide relevant information.

Discussion

1. Trustworthiness and limitations of findings should be considered.

6. PLOS authors have the option to publish the peer review history of their article (what does this mean?). If published, this will include your full peer review and any attached files.

**Do you want your identity to be public for this peer review?** For information about this choice, including consent withdrawal, please see our Privacy Policy.

Reviewer #1: **Yes: **Abdul Jaleel CP

Reviewer #2: No

---

## [Editor Report · Decision Letter 1]

22 Jul 2024

Food taboos and preferences among adolescent girls, pregnant women, breastfeeding mothers, and children aged 6-23 months in Mainland Tanzania: A qualitative study

PGPH-D-24-00652R1

Dear Dr. Masumo,

We are pleased to inform you that your manuscript 'Food taboos and preferences among adolescent girls, pregnant women, breastfeeding mothers, and children aged 6-23 months in Mainland Tanzania: A qualitative study' has been provisionally accepted for publication in PLOS Global Public Health.

Best regards,

Hasanain Faisal Ghazi, phd

Academic Editor